# THCRL: Trusted Hierarchical Contrastive Representation Learning for Multi-View Clustering

## Abstract

Multi-View Clustering (MVC) has garnered increasing attention in recent years. It is capable of partitioning data samples into distinct groups by learning a consensus representation. However, a significant challenge remains: the problem of untrustworthy fusion. This problem primarily arises from two key factors: 1) Existing methods often ignore the presence of inherent noise within individual views; 2) In traditional MVC methods using Contrastive Learning (CL), similarity computations typically rely on different views of the same instance, while neglecting the structural information from nearest neighbors within the same cluster. Consequently, this leads to the wrong direction for multi-view fusion. To address this problem, we present a novel Trusted Hierarchical Contrastive Representation Learning (THCRL). It consists of two key modules. Specifically, we propose the Deep Symmetry Hierarchical Fusion (DSHF) module, which leverages the UNet architecture integrated with multiple denoising mechanisms to achieve trustworthy fusion of multi-view data. Furthermore, we present the Average $K$-Nearest Neighbors Contrastive Learning (AKCL) module to align the fused representation with the view-specific representation. Unlike conventional strategies, AKCL enhances representation similarity among samples belonging to the same cluster, rather than merely focusing on the same sample across views, thereby reinforcing the confidence of the fused representation. Extensive experiments demonstrate that THCRL achieves the state-of-the-art performance in deep MVC tasks.

## 1 Introduction

Driven by accelerating digitalization, data are increasingly generated from multiple heterogeneous sources. For example, autonomous vehicles rely on synchronized multi-camera systems to capture diverse views, enabling real-time decisions for safe navigation. In structural biology, the complex quaternary structures of proteins can be resolved through complementary analytical techniques. Similarly, modern diagnostic practices integrate multi-modal clinical data to enhance diagnostic accuracy. The term "multi-view data" refers to a type of data where each object is described by multiple heterogeneous sources. Multi-View Clustering (MVC) (Chowdhury et al., 2025; Tang et al., 2023) aims to integrate such diverse data to form meaningful clusters. Typical methods utilize view-specific encoders to extract representations from each view, which are then fused into a consensus embedding for subsequent clustering tasks. Nevertheless, significant heterogeneity often exists across views, posing a challenge for effective fusion. To address this issue, various alignment strategies have been proposed. Some methods, for instance, employ KL divergence to align the distribution of labels or representations across views (Hershey & Olsen, 2007). Additionally, Contrastive Learning (CL) has been adopted in MVC to enhance the consistency and compatibility of representations from different views (Zhu et al., 2025).

Although existing methods have considerably advanced the MVC tasks, the problem of untrusted fusion remains unresolved. This problem stems primarily from two factors: 1) Noise in Single or Multiple Views. Since current deep MVC methods (e.g., CoMVC (Trosten et al., 2021), DSIMVC (Tang & Liu, 2022a), DIMVC (Xu et al., 2022a)) commonly rely on basic fusion methods like concatenation or weighted-sum fusion, it becomes challenging to derive a trustworthy fused representation from multi-view data. 2) CL at the Sample Level. CL methods (e.g., MFLVC (Xu et al.,

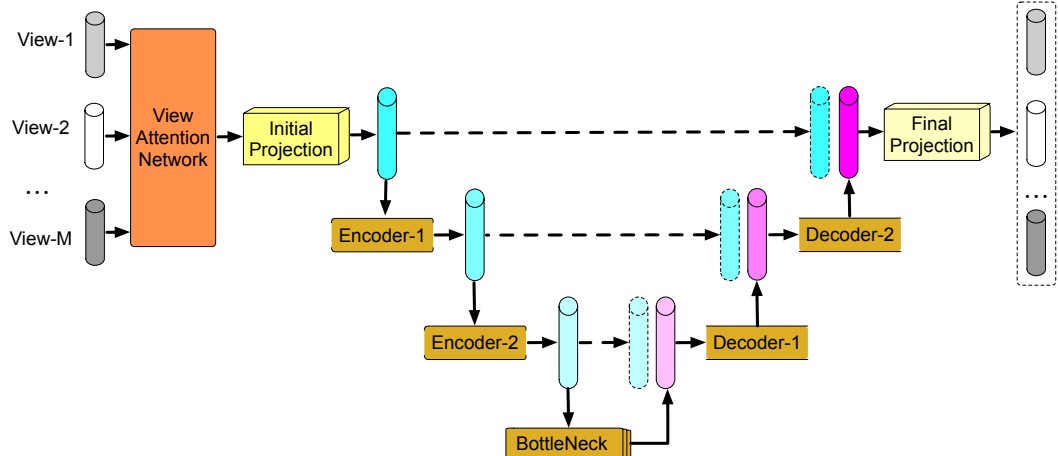

Figure 1: Deep Symmetry Hierarchical Fusion (DSHF). DSHF has multiple denoising mechanisms. First, the View Attention Network performs dynamic weighting on the input features. Second, the Initial Project maps the multi-view feature into a unified feature space. Third, hierarchical fusion achieves denoising features through the UNet architecture with attention networks. Finally, the Final Project reduces the number of feature channels to the original number of views.

2022b), DSIMVC (Tang & Liu, 2022a)) typically take different views of an instance as positive samples. However, these methods can introduce conflicts among samples that belong to the same cluster. Since such samples inherently possess similar representations, the contrastive loss may inadvertently drive the fused representation in an erroneous direction, further compromising the confidence of the fusion process. These two factors collectively degrade the performance of MVC tasks, highlighting the need for more trustworthy fusion.

To address the problem of untrusted fusion, we propose a novel Trusted Hierarchical Contrastive Representation Learning (THCRL). It consists of two core modules, which are Deep Symmetry Hierarchical Fusion (DSHF) and Average $K$-Nearest Neighbors Contrastive Learning (AKCL). As illustrated in Figure 1, DSHF is a new paradigm in deep MVC tasks. This module incorporates three key components: a view attention network, a channel attention network, and a symmetric hierarchical fusion. Together, these components work synergistically to isolate noise patterns in the feature space and achieve trustworthy integration of multi-view data. Furthermore, we present the AKCL module to enhance semantic consistency among samples within the same cluster, rather than merely aligning representations of the same instance across views. This design effectively mitigates a fundamental limitation of conventional CL in MVC, enabling the capture of richer structural semantics and thereby significantly boosting the confidence of the fused representation. The main contributions of this work are summarized as follows:

- We propose a novel DSHF module designed to mitigate noise during the fusion of multi-view data. By incorporating the UNet architecture equipped with multiple denoising mechanisms, DSHF achieves trusted integration of heterogeneous features.

- Unlike previous methods that treat different views of the same instance as positive pairs, we present a new AKCL module. This module enhances representation similarity among samples within the same cluster, thereby significantly improving the confidence of the fused representation.

- Comprehensive experiments demonstrate that the THCRL achieves new state-of-the-art performance in deep MVC tasks, performing better than current methods on the six open datasets.

## 2 RETATED WORK

Spurred by rapid advances in deep learning, recent research has increasingly gravitated toward Deep Multi-View Clustering (MVC) (Yu et al., 2025; Xu et al., 2023). Specifically, these methods utilize deep neural networks to capture the nonlinear feature. Adversarial training is adopted in Deep MVC methods (Li & Liao, 2021; Wang et al., 2022) to align the latent patterns of multiple views and get

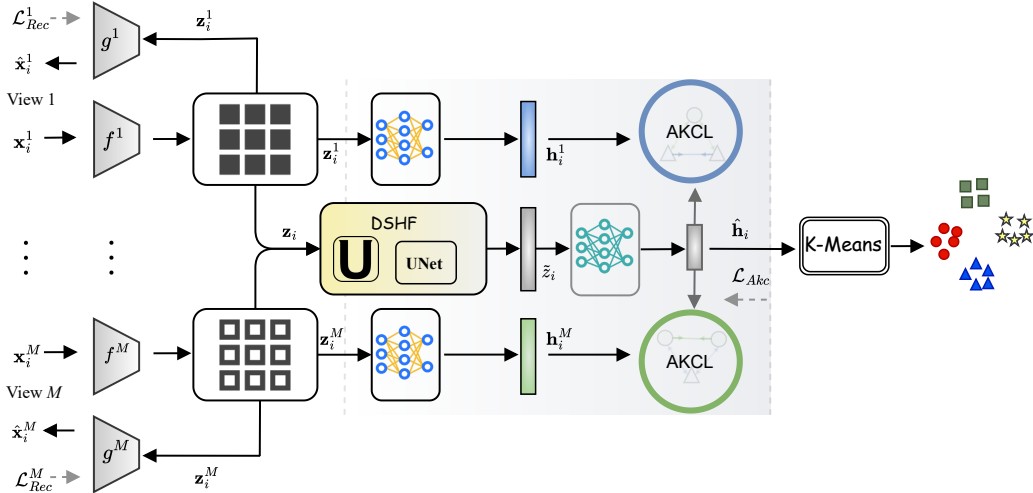

Figure 2: The architecture overview of THCRL. The THCRL method contains two key modules: DSHF and AKCL. DSHF implements trustworthy fusion by the UNet structure equipped with multiple denoising mechanisms. Furthermore, the AKCL module is proposed to enhance representation similarity among samples within the same cluster, as opposed to only concentrating on the consistency at the sample level. It significantly enhances the confidence of the fused representation.

a common representation. Zhou et al. (Zhou & Shen, 2020) employ an attention mechanism to assign an adaptive weight to each view. They then aggregate the view-specific representations into a consensus embedding via a weighted sum determined by those learned weights. Wang et al. (Wang et al., 2022) derive the consensus representation through a weighted aggregation scheme and $l_{1,2}$-norm constraint. Contrastive Learning (CL) can align cross-view representations at the sample level, thereby promoting coherent label distributions. These CL approaches (Yin et al., 2025; Wu et al., 2024b; Guo et al., 2024) have surpassed prior distribution alignment methods in MVC, delivering state-of-the-art performance on a wide range of public benchmarks.

## 3 THE PROPOSED METHODOLOGY

This paper proposes an innovative Trusted Hierarchical Contrastive Representation Learning (THCRL), which aims to address the problem of untrusted fusion in deep Multi-View Clustering (MVC). As illustrated in Figure 2, THCRL comprises two key components: 1) Deep Symmetry Hierarchical Fusion (DSHF); 2) Average $K$-Nearest Neighbors Contrastive Learning (AKCL). The multi-view data, which consists of $N$ samples with $M$ views, is denoted as $\{\mathbf{X}^m = \{x_1^m; ...; x_N^m\} \in \mathbb{R}^{N \times D_m}\}_{m=1}^M$, where $D_m$ represents the feature dimensionality for the $m$-th view.

### 3.1 AUTOENCODER NETWORK

We create the representation of each view by using the Autoencoder Network (Song et al., 2018). For the $m$-th view, $f^m$ represents the encoder module. The encoder module generates the representation of each view as follows:

$$z_i^m = f^m(x_i^m), z_i^m \in \mathbb{R}^{d_\psi}, \tag{1}$$

where $z_i^m$ denotes the representation of the $i$-th instance in the $m$-th view. $d_\psi$ represents the feature size of the $z_i^m$. The $x_i^m$ is restored by the decoder module utilizing the feature $z_i^m$. Let $g^m$ represent the decoder module in the $m$-th view. The recovered sample $\hat{x}_i^m$ is produced by decoding $z_i^m$ in the decoder module:

$$\hat{x}_i^m = g^m(z_i^m). \tag{2}$$

The reconstruction loss is calculated as follows:

$$\mathcal{L}_{\text{Rec}} = \sum_{m=1}^M \sum_{i=1}^N \|x_i^m - g^m(z_i^m)\|_2^2. \tag{3}$$

## 3.2 DEEP SYMMETRY HIERARCHICAL FUSION (DSHF)

We propose a novel Deep Symmetry Hierarchical Fusion (DSHF) method. DSHF is built upon the UNet architecture with multiple denoising mechanisms. DSHF incorporates three denoising components: the View Attention Network, the Channel Attention Network, and the symmetric hierarchical fusion module. DSHF leverages the triple mechanism to effectively disentangle noise patterns in the feature space, enabling the trusted fusion of multi-view data. Its basic constituent units are the Channel Attention Network and the Residual Convolutional Block. In this section, we first introduce these two constituent units and then elaborate on the overall network architecture of DSHF.

### 3.2.1 CHANNEL ATTENTION NETWORK

Given an input tensor $q \in \mathbb{R}^{C \times L}$, $C$ represents the number of channels, and $L$ is the feature dimension. The Channel Attention Network (CAN) computes a channel-wise scaling vector and applies it to the original features. The CAN operation is defined as follows:

$$\hat{q} = CAN(q) = q \odot mlp^1(\frac{1}{L}\sum_{l=1}^{L} q_{[:,l]}), \tag{4}$$

where $\odot$ denotes the Hadamard product. $mlp^1$ represents a multi-layer perceptron. Its function is to suppress noise channels and enhance effective features.

### 3.2.2 RESIDUAL CONVOLUTIONAL BLOCK

The Residual Convolutional Block (RCBlock) integrates a convolutional network, a channel attention network, and a residual connection. Given an input tensor $o \in \mathbb{R}^{C \times L}$, $C$ represents the number of channels, and $L$ is the feature dimension. The block performs the following operations:

$$\hat{\mathbf{o}} = RCBlock(o) = CAN(conv1d(conv1d(o))) + conv1d(o), \tag{5}$$

where $conv1d$ denotes a one-dimensional convolutional neural network.

### 3.2.3 OVERALL NETWORK ARCHITECTURE OF DSHF

We concatenate the vectors $\{z_i^m\}_{m=1}^{M}$ of $M$ views as follows:

$$z_i = [z_i^1; z_i^2; \ldots; z_i^M]^T, \ z_i \in \mathbb{R}^{M \times d_\psi}. \tag{6}$$

$M \times d_\psi$ represents the $M$ rows and $d_\psi$ columns of the matrix. When processing data, we consider $M$ view features as $M$ channel data. $T$ represents the transpose of the matrix.

**View Attention Network.** The View Attention Network aims to learn view-specific importance weights. This operation is defined as described below:

$$z_i^a = z_i \odot w^a, \ z_i^a \in \mathbb{R}^{M \times d_\psi}, \tag{7}$$

where $w^a \in \mathbb{R}^M$ is the learnable weight vector for views. Its function is to automatically learn the confidence weights of each view and reduce the impact of noisy views.

**Initial Projection.** The Initial Projection uses a one-dimensional convolutional neural network to transform $z_i^a$ to $z_i'$ as follows:

$$z_i' = conv1d(z_i^a), \ z_i' \in \mathbb{R}^{C_0 \times d_\psi}, \tag{8}$$

where $C_0$ denotes the base channel dimension.

**Encoder Pathway.** The Encoder Pathway is a hierarchical feature denoising by $U$ encoder stages. At stage $u \in [0, 1, \ldots, U-1]$:

$$p_i^{(u+1)} = RCBlock(z_i^{(u)}), \ p_i^{(u+1)} \in \mathbb{R}^{C_{u+1} \times L_u}, \tag{9}$$

$$z_i^{(u+1)} = MaxPool1D(p_i^{(u+1)}), \ z_i^{(u+1)} \in \mathbb{R}^{C_{u+1} \times L_{u+1}}, \tag{10}$$

where $z_i^{(0)} = z_i'$, $L_u = d_\psi/2^u$, and $C_u = 2^u C_0$. $p_i^{(u+1)}$ is stored as skip connection. $MaxPool1D$ denotes a one-dimensional max pooling operator. The Encoder Pathway captures more robust features through $U$ encoder stages. These $U$ stages have the function of continuously removing noise.

**Bottleneck Layer.** The Bottleneck Layer utilizes the RCBlock module for deepest feature processing. Its processing procedure is as follows:

$$z_i^{(b)} = \text{RCBlock}(z_i^{(U)}), \ z_i^{(b)} \in \mathbb{R}^{C_U \times L_U}, \tag{11}$$

where $L_U = d_\psi/2^U$, $C_U = 2^U C_0$. It performs semantic extraction to capture the most essential global feature representation while simultaneously compressing spatial dimensions and eliminating noisy data.

**Decoder Pathway.** The Decoder Pathway is a multi-stage feature reconstruction with long-range information fusion. It contains $U$ steps. At step $u \in [0, 1, \ldots, U-1]$:

$$e_i^{(u+1)} = UpConv(t_i^{(u)}), \ e_i^{(u+1)} \in \mathbb{R}^{C'_{u+1} \times L'_{u+1}}, \tag{12}$$

$$\xi_i^{(u+1)} = cat(e_i^{(u+1)}, p_i^{(U-u)}), \ \xi_i^{(u+1)} \in \mathbb{R}^{C''_{u+1} \times L'_{u+1}}, \tag{13}$$

$$t_i^{(u+1)} = RCBlock(\xi_i^{(u+1)}), \ t_i^{(u+1)} \in \mathbb{R}^{C'_{u+1} \times L'_{u+1}}, \tag{14}$$

where $t_i^0 = z_i^{(b)}$, $L'_u = d_\psi/2^{U-u}$, $C'_u = 2^{U-u}C_0$, and $C''_u = 2^{U-u}C_0 + 2^{U-u+1}C_0$. $UpConv$ is Transposed convolution. $cat$ represents the concatenation operator of two vectors along the channel dimension.

**Final Projection.** The Final Projection reduces the number of channels from $C_0$ to $M$ as described below:

$$\hat{z}_i = conv1d(z'_i + t_i^{(U)}), \ \hat{z}_i \in \mathbb{R}^{M \times d_\psi}. \tag{15}$$

**Flattened Operator.** The sequence vector $\hat{z}_i$ is expanded into a one-dimensional vector as follows:

$$\tilde{z}_i = re(\hat{z}_i), \ \tilde{z}_i \in \mathbb{R}^{M d_\psi}, \tag{16}$$

where $re$ represents the vector deformation operation.

### 3.3 Average $K$-Nearest Neighbors Contrastive Learning (AKCL)

This paper presents the Average $K$-Nearest Neighbors Contrastive Learning (AKCL) to enhance representation similarity among samples belonging to the same cluster. We construct the adjacency matrix $S^m$ leveraging the $K$-Nearest Neighbors graph (Peterson, 2009) as follows:

$$S_{ij}^m = \begin{cases} 1 & j \in \delta_i^m \\ 0 & \text{otherwise} \end{cases}. \tag{17}$$

In the $m$-th view, $\delta_i^m$ represents the set of $K$ nearest neighbors of $z_i^m$ according to the Euclidean distance. By summing up the adjacency matrices of $M$ views and calculating the average, we get

$$S_{ij} = \frac{1}{M} \sum_{m=1}^M S_{ij}^m. \tag{18}$$

AKCL needs to align the dimensions of each view feature and the fused feature. The precise computation is as follows:

$$\hat{h}_i = mlp^2(\tilde{z}_i), \ \hat{h}_i \in \mathbb{R}^{d_\phi}, \tag{19}$$

where the $mlp^2$ operator is used to reduce the dimension of $\tilde{z}_i$. $d_\phi$ represents the feature dimension of the $\hat{h}_i$. Likewise, we use the $mlp^{3,m}$ operator to decrease the dimension on each view feature $z_i^m$,

$$h_i^m = mlp^{3,m}(z_i^m), \ h_i^m \in \mathbb{R}^{d_\phi}. \tag{20}$$

For the $h_i^m$, $d_\phi$ denotes the feature dimension. The similarity between the view-specific embedding $h_i^m$ and the common embedding $\hat{h}_i$ is calculated using the cosine function:

$$C\left(\hat{h}_i, h_i^m\right) = \cos(\hat{h}_i, h_i^m). \tag{21}$$

The loss of the AKCL is computed as follows:

$$\mathcal{L}_{\text{Akc}} = -\frac{1}{2N} \sum_{i=1}^N \sum_{m=1}^M \log \frac{e^{C(\hat{h}_i, h_i^m)/\tau}}{\sum_{j=1}^N e^{(1-S_{ij}) C(\hat{h}_i, h_j^m)/\tau} - e^{1/\tau}}, \tag{22}$$

Table 1: Summary of the six public datasets.

| Datasets | Samples | Views | Clusters | #-view Dimensions |
|----------|---------|-------|----------|-------------------|
| MNIST | 60000 | 3 | 10 | [342, 1024, 64] |
| OutScene | 2688 | 4 | 8 | [512, 432, 256, 48] |
| BRCA | 398 | 4 | 4 | [2000, 2000, 278, 212] |
| Hdigit | 10000 | 2 | 10 | [784, 256] |
| Synthetic3d | 600 | 3 | 3 | [3, 3, 3] |
| LandUse | 2100 | 3 | 21 | [20, 59, 59] |

where $\tau$ is the temperature factor. According to Eq. (18), $S_{ij}$ is generated. To increase the similarity of the samples in the cluster, we add $S_{ij}$ to Eq. (22). The two samples do not belong to the same cluster when $S_{ij}$ is small. Currently, the $(1 - S_{ij}) \, \mathrm{C}(\hat{h}_i, h_j^m)$ is important. A significantly large $S_{ij}$ implies that the two samples are part of the same cluster. At present, the impact of $(1 - S_{ij}) \, \mathrm{C}(\hat{h}_i, h_j^m)$ is nearly negligible.

The definition of the total loss function is as follows:

$$\mathcal{L} = \mathcal{L}_{\mathrm{Rec}} + \lambda \mathcal{L}_{\mathrm{Akc}}, \tag{23}$$

where $\lambda$ balances the impact of $\mathcal{L}_{\mathrm{Rec}}$ and $\mathcal{L}_{\mathrm{Akc}}$.

### 3.4 CLUSTERING MODULE

We employ the $K$-Means algorithm as the clustering function (MacKay et al., 2003; Yan et al., 2023). The fused embedding $\mathbf{H} = \{\hat{h}_1, ..., \hat{h}_N\}$ serves as the input features to the $K$-Means method. To evaluate the performance of the learned representations, the number of clusters in the clustering algorithm is set to correspond to the true number of classes in the dataset, thereby enabling the computation of standard clustering metrics.

## 4 EXPERIMENTS

We perform comprehensive experiments to evaluate the Trusted Hierarchical Contrastive Representation Learning (THCRL) method on six public datasets for deep Multi-View Clustering (MVC) tasks. To demonstrate its advantage, we compare it against eight state-of-the-art methods across three metrics. In addition, we conduct analyses on hyperparameters, visualization, and convergence.

### 4.1 BENCHMARK DATASETS

As shown in Table 1, we use six open datasets with different scales for evaluation (Yan et al., 2023). The six datasets include Hdigit, Synthetic3d, BRCA, MNIST, OutScene, and LandUse. The performance of MVC is regularly evaluated using the six public datasets.

### 4.2 COMPARED METHODS

We compare the effectiveness of THCRL with eight current state-of-the-art MVC methods. These baselines are all deep learning methods (including DEMVC (Xu et al., 2021), DSMVC (Tang & Liu, 2022b), DealMVC (Yang et al., 2023), GCFAggMVC (Yan et al., 2023), SCMVC (Wu et al., 2024a), MVCAN (Xu et al., 2024), ACCMVC (Yan et al., 2024), and DMAC (Wang et al., 2025)).

### 4.3 IMPLEMENTATION DETAILS

The PyTorch framework is used for developing the source code of THCRL. The dropout parameter of the network is set to $0.1$. We set the batch size $b$ to 256 during the model training procedure. There are two phases of THCRL training: pre-training and fine-tuning. The number of pre-training epochs $T_p$ is 200, and the number of fine-tuning epochs $T_f$ is 200. The temperature coefficient $\tau$ is set to $0.5$. We unify the dimensions of the vectors outputted by the encoder and the input vectors for the contrastive loss, making $d_\psi$ equal to 512 and $d_\phi$ equal to 128. We set the depth $U$ of DSHF to 4.

The parameter $K$ of the AKCL module is 10. The learning rate of deep neural network is 0.0003. The balance parameter $\lambda$ of the two loss functions is set to 1. The hardware platforms we utilized in our experiment are Intel(R) Xeon(R) Platinum 8358 CPU and Nvidia V100 GPU.

Table 2: Clustering result comparison on the MNIST, OutScene, and BRCA datasets. The best results in bold, second in underline.

| Methods | MNIST | | | OutScene | | | BRCA | | |
|---|---|---|---|---|---|---|---|---|---|
| | ACC | NMI | PUR | ACC | NMI | PUR | ACC | NMI | PUR |
| DEMVC [IS'21] | 0.9734 | 0.9587 | 0.9734 | 0.3532 | 0.3049 | 0.3553 | 0.3945 | 0.0619 | 0.4397 |
| DSMVC [CVPR'22] | 0.9671 | 0.9215 | 0.9671 | 0.4702 | 0.3455 | 0.4747 | 0.5101 | 0.2300 | 0.5704 |
| DealMVC [MM'23] | 0.9804 | 0.9593 | 0.9804 | 0.5874 | 0.5111 | 0.5885 | 0.6256 | 0.3721 | 0.6357 |
| GCFAggMVC [CVPR'23] | 0.6949 | 0.6768 | 0.6949 | 0.3594 | 0.3141 | 0.3891 | 0.3392 | 0.0363 | 0.4296 |
| SCMVC [TMM'24] | 0.8610 | 0.8263 | 0.8610 | 0.5893 | 0.4645 | 0.5893 | 0.5603 | 0.2431 | 0.6055 |
| MVCAN [CVPR'24] | 0.9863 | 0.9612 | 0.9863 | 0.7072 | 0.5690 | 0.7072 | 0.5603 | 0.3288 | 0.6130 |
| ACCMVC [TNNLS'24] | 0.9886 | 0.9659 | 0.9886 | 0.6347 | 0.5161 | 0.6347 | 0.4673 | 0.1898 | 0.5704 |
| DMAC [AAAI'25] | 0.9720 | 0.9302 | 0.9720 | 0.4349 | 0.3460 | 0.4661 | 0.5327 | 0.3322 | 0.6030 |
| THCRL (Ours) | **0.9920** | **0.9747** | **0.9920** | **0.7474** | **0.5910** | **0.7474** | **0.6332** | **0.3820** | **0.6608** |

Table 3: Clustering result comparison on the Hdigit, Synthetic3d, and LandUse datasets. The best in bold, the second underlined.

| Methods | Hdigit | | | Synthetic3d | | | LandUse | | |
|---|---|---|---|---|---|---|---|---|---|
| | ACC | NMI | PUR | ACC | NMI | PUR | ACC | NMI | PUR |
| DEMVC [IS'21] | 0.4318 | 0.3407 | 0.4318 | 0.8267 | 0.6373 | 0.8267 | 0.2057 | 0.2032 | 0.2171 |
| DSMVC [CVPR'22] | 0.9669 | 0.9368 | 0.9669 | 0.9567 | 0.8267 | 0.9567 | 0.2695 | 0.3476 | 0.3048 |
| DealMVC [MM'23] | 0.1980 | 0.1449 | 0.1982 | 0.8983 | 0.6904 | 0.8983 | 0.1895 | 0.2014 | 0.1929 |
| GCFAggMVC [CVPR'23] | 0.9730 | 0.9272 | 0.9730 | 0.5750 | 0.3462 | 0.6283 | 0.2610 | 0.2991 | 0.2833 |
| SCMVC [TMM'24] | 0.9402 | 0.8629 | 0.9402 | 0.9417 | 0.7944 | 0.9417 | 0.2595 | 0.2791 | 0.2752 |
| MVCAN [CVPR'24] | 0.9596 | 0.9047 | 0.9596 | 0.9850 | 0.9262 | 0.9850 | 0.2395 | 0.3099 | 0.2867 |
| ACCMVC [TNNLS'24] | 0.9706 | 0.9220 | 0.9706 | 0.9600 | 0.8381 | 0.9600 | 0.2610 | 0.2952 | 0.2767 |
| DMAC [AAAI'25] | 0.9113 | 0.8140 | 0.9113 | 0.6083 | 0.2593 | 0.6083 | 0.2429 | 0.2899 | 0.2786 |
| THCRL (Ours) | **0.9970** | **0.9903** | **0.9970** | **0.9883** | **0.9395** | **0.9883** | **0.2943** | **0.3497** | **0.3100** |

## 4.4 EVALUATION METRICS

We utilize three standard quantitative metrics, including unsupervised clustering accuracy (ACC), normalized mutual information (NMI), and purity (PUR). The larger these three indicators are, the better the clustering performance of the model.

## 4.5 EXPERIMENTAL COMPARATIVE RESULTS

As illustrated in Table 2 and Table 3, THCRL outperforms eight state-of-the-art methods (including DEMVC, DSMVC, DealMVC, GCFAggMVC, SCMVC, MVCAN, ACCMVC, and DMAC) in six benchmark datasets. Specifically, we obtain the following results: On the OutScene dataset, THCRL surpasses the second-best method (MVCAN) by 4.02% in ACC. Similarly, on the Hdigit dataset, our method outperforms GCFAggMVC by 2.40% in ACC, with consistently improved NMI and PUR. In the other four datasets, THCRL furthermore achieves state-of-the-art results and exhibits strong generalizability.

## 4.6 ABLATION STUDY

We perform an ablation experiment to evaluate the two key components of THCRL: Deep Symmetry Hierarchical Fusion (DSHF) and Average $K$-Nearest Neighbors Contrastive Learning (AKCL).

**Effectiveness of DSHF module.** The "w/o DSHF" indicates eliminating the DSHF module from the THCRL framework. All view-specific representations $\{z_i^m\}_{m=1}^M$ are concatenated to form the fused representation $z_i$. As shown in Table 4, the results of w/o DSHF are 0.23, 6.25, 2.52, 2.37, 2.34, and 2.53 percent lower than those of the THCRL in the ACC term. The results show that DSHF implements the trusted fusion, considerably enhancing the accuracy of deep MVC tasks.

**Validity of AKCL module.** The "w/o AKCL" denotes the removal of the AKCL module from the THCRL framework. Table 4 shows that the results of w/o AKCL are 53.44, 34.45, 25.88, 29.79,

Table 4: Ablation study on the six common datasets.

| Datasets | Method | ACC | NMI | PUR |
|---|---|---|---|---|
| MNIST | w/o DSHF | 0.9897 | 0.9692 | 0.9897 |
| | w/o AKCL | 0.4576 | 0.5155 | 0.5279 |
| | THCRL | **0.9920** | **0.9747** | **0.9920** |
| OutScene | w/o DSHF | 0.6849 | 0.5502 | 0.6849 |
| | w/o AKCL | 0.4029 | 0.2793 | 0.4267 |
| | THCRL | **0.7474** | **0.5910** | **0.7474** |
| BRCA | w/o DSHF | 0.6080 | 0.3295 | 0.6332 |
| | w/o AKCL | 0.3744 | 0.0958 | 0.5075 |
| | THCRL | **0.6332** | **0.3820** | **0.6608** |
| Hdigit | w/o DSHF | 0.9733 | 0.9288 | 0.9733 |
| | w/o AKCL | 0.6991 | 0.5049 | 0.6991 |
| | THCRL | **0.9970** | **0.9903** | **0.9970** |
| Synthetic3d | w/o DSHF | 0.9583 | 0.8352 | 0.9583 |
| | w/o AKCL | 0.9633 | 0.8625 | 0.9633 |
| | THCRL | **0.9883** | **0.9395** | **0.9883** |
| LandUse | w/o DSHF | 0.2690 | 0.3237 | 0.2833 |
| | w/o AKCL | 0.1938 | 0.2562 | 0.2348 |
| | THCRL | **0.2943** | **0.3497** | **0.3100** |

1.84, and 10.05 percent inferior to those of the THCRL on the ACC term. AKCL improves representation similarity among samples within the same cluster, rather than merely aligning representations of the same instance across views. Consequently, it boosts the results in MVC tasks.

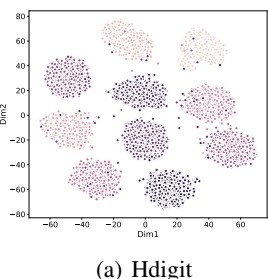
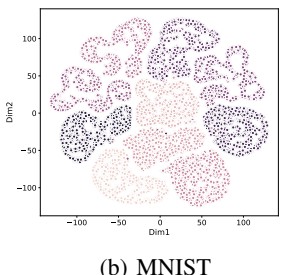
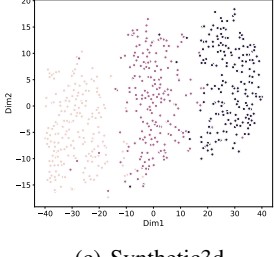

| (a) Hdigit | (b) MNIST | (c) Synthetic3d |

Figure 3: The visualization results of the fused representations $\{\hat{h}_i\}_{i=1}^N$ on the Hdigit, MNIST, and Synthetic3d datasets after convergence.

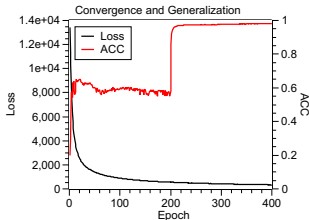
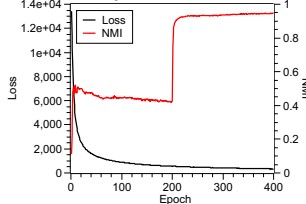
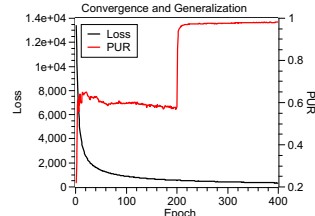

Figure 4: The convergence analysis on the Hdigit dataset. In the figure, the test ACC, NMI, and PUR are shown at the top, and the training loss is depicted at the bottom.

## 4.7 VISUALIZATION

As illustrated in Figure 3, we employ the t-SNE approach (Van der Maaten & Hinton, 2008) to visualize the fused representations $\{\hat{h}_i\}_{i=1}^N$. We run visualization experiments using the three datasets:

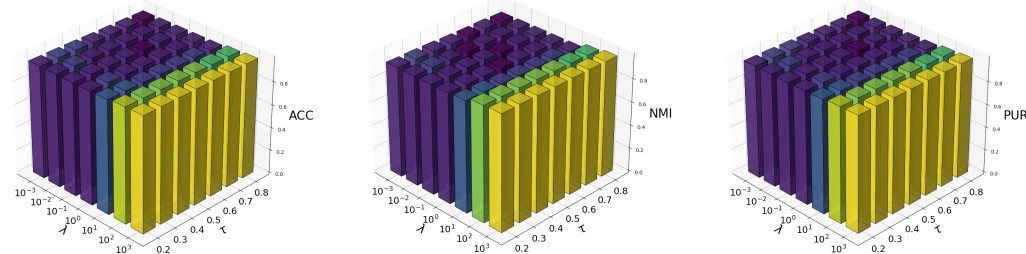

Figure 5: The hyperparameter analysis on the Hdigit dataset. The figure shows the changes in three evaluation metrics: ACC, NMI, and PUR. The metrics are influenced by two hyperparameters $\lambda$ and $\tau$. $\lambda$ is the combination coefficient of two loss functions. $\tau$ denotes the temperature coefficient.

Hdigit, MNIST, and Synthetic3d. Figure 3 shows the visualization results for the Hdigit, MNIST, and Synthetic3d datasets as $(a)$, $(b)$, and $(c)$, respectively. The clustering findings show that the boundaries are quite obvious and the clusters are fully separated after convergence. There is obviously no interrelated between clusters. The visualization of Figure 3(a) demonstrates that the samples are divided into 10 groups, and the visualization effect is excellent, which can also be correlated with the 10 labels of the Hdigit dataset. Likewise, the visualization results of Figures 3(b) and 3(c) display 10 and 3 groups, which are in line with the labels of the MNIST and Synthetic3d datasets.

### 4.8 CONVERGENCE ANALYSIS AND GENERALIZATION ABILITY

We perform experiments on the Hdigit dataset to verify the convergence and generalization efficacy of THCRL. In Figure 4, the test ACC, NMI, PUR, and training loss are depicted. The three subfigures that make up Figure 4 show the variation curves for test ACC, NMI, PUR, and training loss. As training goes on, the loss curve gradually drops until it stabilizes at 300 epochs. This shows that the deep MVC network is working well. At the beginning of the experiment, the test ACC, NMI, and PUR for the evaluation measure exhibit a notable increase. After 400 epochs, the ACC, NMI, and PUR hold steady and remain unchanged with further training, demonstrating strong generalization and the absence of overfitting.

### 4.9 PARAMETER ANALYSIS

As illustrated in Figure 5, we make hyperparameter analysis experiments. In particular, we analyze the effect of two hyperparameters, $\lambda$ and $\tau$, on clustering evaluation metrics. $\lambda$ denotes the combination coefficient of two loss functions. $\tau$ is the temperature coefficient. The test ACC, NMI, and PUR serve as the evaluation metrics. We run hyperparameter experiments on the Hdigit dataset. The value of $\lambda$ ranges from $10^{-3}$ to $10^3$. The values of $\tau$ are 0.2, 0.3, 0.4, 0.5, 0.6, 0.7, and 0.8. We depict three subfigures that show the changes in the three metrics ACC, NMI, and PUR under the condition of hyperparameters. Figure 5 indicates that ACC, NMI, and PUR are not considerably impacted by the two hyperparameters and exhibit rather steady performance.

## 5 CONCLUSION AND FUTURE WORK

This paper proposes a novel Trusted Hierarchical Contrastive Representation Learning (THCRL) method. It aims to implement trusted fusion in Multi-View Clustering (MVC). It consists of two key modules: Deep Symmetry Hierarchical Fusion (DSHF) and Average $K$-Nearest Neighbors Contrastive Learning (AKCL). The DSHF module, which utilizes the UNet framework with multiple denoising mechanisms, enables the trustworthy fusion of multi-view data. Meanwhile, the AKCL module alleviates conflicts among samples within the same cluster. It further enhances the confidence of the fused representation. Extensive experiments demonstrate that THCRL outperforms state-of-the-art methods in deep MVC tasks. Specifically, the efficacy of THCRL is validated through comparisons with eight baseline methods on six publicly available datasets. In summary, THCRL provides an effective representation learning framework with strong potential for practical applications.

## ETHICS STATEMENT

This work adheres to the highest standards of academic and research ethics. Our research does not involve human subjects, the collection or use of personal or sensitive data, and presents no foreseeable risks to the environment, society, safety, or welfare. All datasets utilized in this study are publicly available. The research was conducted with a responsible approach, and we have diligently considered the potential limitations and broader impacts of our work. The authors declare no competing interests, whether financial or non-financial.

## REPRODUCIBILITY

To facilitate the reproducibility of our work, we are committed to providing comprehensive details and resources. In the additional materials, we provide the code, dataset, and startup script. All publicly available datasets used in our experiments are explicitly cited in the main text. The experimental section in the paper provides a detailed description of the model architectures, hyperparameters (e.g., learning rate, batch size), and the computational environment (e.g., framework versions, GPU specifications).

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

# A APPENDIX

## A.1 THE USE OF LARGE LANGUAGE MODELS (LLMS)

In this work, we do not use the Large Language Models (LLMs) as an assistant.

