# OpenReview forum: "THCRL: Trusted Hierarchical Contrastive Representation Learning for Multi-View Clustering"
_ICLR.cc/2026/Conference — ICLR 2026 Conference Withdrawn Submission_

### Official Review · Reviewer_3WAe · 2025-10-24

**Soundness:** 3
**Presentation:** 2
**Contribution:** 1
**Rating:** 2
**Confidence:** 4

**Summary:**

This paper proposes a new framework called THCRL (Trusted Hierarchical Contrastive Representation Learning) to address the "untrusted fusion" problem in multi-view clustering (MVC). The paper points out two major flaws in current deep MVC methods: first, the multi-view fusion process is easily corrupted by single-view noise, resulting in unreliable global representations; second, contrastive learning is typically based on sample-level cross-view matching, which ignores the structural information of neighboring samples in the same cluster, thereby weakening cluster consistency. To address these issues, THCRL designs two core modules: Deep Symmetry Hierarchical Fusion (DSHF) and Average K-Nearest Neighbors Contrastive Learning (AKCL). DSHF, based on the UNet architecture, introduces multiple denoising mechanisms, including view attention, channel attention, and residual convolutional blocks, to achieve high-confidence feature fusion. AKCL constructs an intra-cluster adjacency matrix based on the K-nearest neighbor graph and enforces cross-view consistency at the cluster level using an average adjacency-constrained contrastive loss.

**Strengths:**

1. The DSHF module, through mechanisms such as the view attention network, channel attention network, and symmetric hierarchical fusion, effectively isolates and suppresses noise within the feature space. This achieves a more reliable fusion of multi-view data compared to traditional concatenation or weighted-sum approaches.
2. The core concept of the AKCL module is to enhance the representation similarity among samples within the same cluster, rather than merely focusing on different views of the same instance. This design better captures the intrinsic clustering structure of the data, thereby significantly improving the confidence of the fused representation.
3. This paper benchmarks THCRL against eight state-of-the-art MVC methods on six benchmark datasets, demonstrating its superiority and generalization ability.

**Weaknesses:**

1. There is a significant overlap between this manuscript and "Trusted Mamba Contrastive Network for Multi-View Clustering" (Zhu et al., 2025), " Self-supervised Trusted Contrastive Multi-view Clustering with Uncertainty Refined" (Hu et al., 2025) in terms of motivation, proposed methodology (e.g., loss functions, model architecture, block diagrams), and the logical structure of the text, particularly in the Abstract and Introduction sections. The current work merely cites the aforementioned paper without providing a detailed comparison or explicitly stating which components were adopted. This lack of clear differentiation raises concerns regarding potential duplicate submission or plagiarism. The above two papers should be included and discussed with their relations.
2. The manuscript fails to specify how the MNIST dataset, which is conventionally single-view, was adapted for a multi-view setting. Furthermore, given that MNIST is a relatively basic dataset, its use is insufficient to demonstrate the advanced capabilities and novelty of the proposed method.
3. The bibliography lacks in-text cross-referencing, making it difficult to trace citations.
4. The writing in Section 4 is fragmented; each subsection consists of only a few sentences, which prevents a thorough and in-depth discussion.
5. MInor issues: Self-weighted contrastive fusion for deep multi-view clustering, this paper has been cited for twice in the references. This is a mistake.

**Questions:**

1. What is the rationale for employing the UNet architecture and the k-nearest neighbors algorithm? Compared to the structurally similar “Trusted Mamba Contrastive Network for Multi-View Clustering” (TMCN), which utilizes Mamba and cosine similarity, what are the distinct advantages of these modifications?
2. Does performing contrastive learning on the neighboring structure within the same cluster introduce a potential risk of incorporating false positive pairs, especially in cases where the initial neighborhood graph may be imperfect?
3. The parameter sensitivity analysis was conducted on only a single dataset. Such limited results are insufficient to substantiate the conclusion that the model is parameter-insensitive. Please provide more comprehensive experimental results across multiple datasets to demonstrate robustness.
4. Please elucidate the novel contributions of this paper in comparison to the "Trusted Mamba Contrastive Network for Multi-View Clustering."

---

### Official Review · Reviewer_PKer · 2025-10-31

**Soundness:** 3
**Presentation:** 3
**Contribution:** 3
**Rating:** 6
**Confidence:** 4

**Summary:**

To address the untrustworthy fusion in Multi-View Clustering (MVC), this paper proposes Trusted Hierarchical Contrastive Representation Learning (THCRL), comprising Deep Symmetry Hierarchical Fusion (DSHF) and Average K-Nearest Neighbors Contrastive Learning (AKCL). DSHF leverages a UNet architecture with denoising mechanisms to achieve trustworthy multi-view integration. AKCL enhances semantic consistency among same-cluster samples by aligning fused and view-specific representations via K-nearest neighbors, mitigating limitations of conventional sample-level contrastive learning. Extensive experiments on six datasets demonstrate THCRL outperforms eight SOTA methods, validating its effectiveness and generalizability in deep MVC tasks.

**Strengths:**

1.	This framework introduces a UNet-based Deep Symmetry Hierarchical Fusion with multi-denoising components, effectively isolating noise to enhance fusion trustworthiness.
2.	The experiment demonstrates state-of-the-art clustering results on six datasets, outperforming eight existing SOTA methods and validating strong generalizability.
3.	This study conducts comprehensive ablation experiments, confirming the necessity of DSHF and AKCL modules through significant performance drops when either component is removed.

**Weaknesses:**

1.	This paper only analyzes the impact of loss weight and temperature, lacking systematic exploration of other key parameters like encoder stages or neighbor count.
2.	UNet-based DSHF may increase computational complexity, but no comparison with lightweight MVC baselines is provided.
3.	Although DSHF claims denoising capabilities, this paper does not include controlled denoising experiments to quantify performance under varying noise levels.

**Questions:**

Please refer to the Weaknesses.

---

### Official Review · Reviewer_MSsB · 2025-10-31

**Soundness:** 2
**Presentation:** 2
**Contribution:** 2
**Rating:** 2
**Confidence:** 5

**Summary:**

In this manuscript, the authors address the untrusted fusion problem caused by view noise and ignoring nearest neighbors within the same cluster. They propose a Trusted Hierarchical Contrastive Representation Learning method consisting of two modules: Deep Symmetry Hierarchical Fusion and Deep Symmetry Hierarchical Fusion. The proposed THCRL model significantly improves the confidence of the fused representation and achieves superior clustering performance in deep MVC tasks.

**Strengths:**

1. This paper has a clear organizational structure that facilitates reading, and provides thorough descriptions of both the motivation and experimental results.
2. The comparative experimental results in Tables 2 and 3 are comprehensive, and the ablation study results in Table 4 demonstrate that both modules contribute positively to improving clustering performance.

**Weaknesses:**

1. The paper's motivation and proposed methods lack innovation. The issue of view noise was already addressed in "Investigating and mitigating the side effects of noisy views for self-supervised clustering algorithms in practical multi-view scenarios" (2024 CVPR). The structural information of nearest neighbors within the same cluster was discussed in "Twin Contrastive Learning for Online Clustering" (2022 IJCV). Furthermore, the problem of untrusted fusion in unsupervised scenarios has been extensively studied in recent years.
2. The mathematical notation in the paper's methods section is inconsistent. Vectors are sometimes bold and sometimes italicized; matrices are sometimes represented by set symbols and sometimes italicized; network structures are sometimes represented by bold and sometimes italicized. This includes, but is not limited to, formulas 5/11/14, etc. For a conference on ICLR's caliber, such extensive errors raise questions about the overall scholarly rigor and meticulousness of the research, making it difficult to have full confidence in the technical details.
3. Except for the MNIST and Hdigit datasets, the other datasets are too small to ensure experiments are convincing and solid. Furthermore, MNIST and Hdigit are both handwritten digit datasets, and their classification is too simplistic, making it difficult to ensure the experiments are convincing.

**Questions:**

Please see the Paper Weaknesses.

---

### Official Review · Reviewer_JJLL · 2025-11-01

**Soundness:** 3
**Presentation:** 3
**Contribution:** 2
**Rating:** 2
**Confidence:** 4

**Summary:**

This paper proposes a novel multi-view clustering method aimed at addressing the problem of unreliable data fusion. The paper is clearly structured, and the proposed method achieves certain results compared to some state-of-the-art algorithms. However, this work does not appear to make a significant contribution to the forefront of the research field. First, the paper’s research in the field is insufficient, resulting in a failure to grasp the latest and most effective problems, or to offer a completely novel approach to previously well-known issues. Second, the effectiveness of the module is questionable. Although it surpasses the methods mentioned in the paper, experiments show that the carefully designed DSHF does not demonstrate real effectiveness, and a lack of more detailed ablation experiments to prove this is a significant deficiency.

**Strengths:**

1.This article is clearly structured, clearly expressed, and easy to understand.
2.The method proposed in this paper is an improvement over some existing methods.

**Weaknesses:**

1.The two key factors claimed in the paper, 1. the intrinsic noise of individual views, and 2. CL at the Sample Level, have been addressed in numerous papers. The authors clearly did not carefully investigate the latest progress in this field. For example, regarding the second key factor, please refer to the following papers. The problem of false negatives has been considered in a large number of works, but the paper does not analyze or mention the progress.
[1] Li, Junnan, et al. “Prototypical Contrastive Learning of Unsupervised Representations.” International Conference on Learning Representations.
[2] Dong, Shihao, et al. “Center-Oriented Prototype Contrastive Clustering.” arXiv preprint arXiv:2508.15231 (2025).
[3] Pan, Erlin, and Zhao Kang. “Multi-view contrastive graph clustering.” Advances in neural information processing systems 34 (2021): 2148-2159.
2.The author claims that the key problem of untrustworthy fusion has also been the subject of extensive research. The paper also fails to introduce relevant research. For example, [1] Geng, Yu, et al. “Uncertainty-aware multi-view representation learning.” Proceedings of the AAAI Conference on Artificial Intelligence. Vol. 35. No. 9. 2021. A detailed survey of the MVC field, including the most relevant work on the key issues and technologies in the text, should be reviewed and cited.
3.Ablation experiments confirm that AKCL is the core module of THCRL. Despite its complex design, DSHF does not deliver the expected results, being only slightly better than directly concatenating multiple views of z. If z were fused even slightly more effectively, DSHF might not even be useful. The paper mentions DSHF’s three denoising components several times, but there’s no intuitive justification. As a core component of the paper, DSHF needs to be clearly stated as necessary, not just providing a marginal performance improvement.

**Questions:**

1.Is DSHF necessary to address key factor 1?
2. Others questions please see above.

---

### Note · Authors · 2025-11-12

I have read and agree with the venue's withdrawal policy on behalf of myself and my co-authors.